# Structure of the mitoribosomal small subunit with streptomycin reveals Fe-S clusters and physiological molecules

**Yuzuru Itoh**[1†], **Vivek Singh**[1†], **Anas Khawaja**[2,3†], **Andreas Naschberger**[1], **Minh Duc Nguyen**[2,3], **Joanna Rorbach**[2,3*], **Alexey Amunts**[1*]

[1]Science for Life Laboratory, Department of Biochemistry and Biophysics, Stockholm University, Stockholm, Sweden; [2]Department of Medical Biochemistry and Biophysics, Karolinska Institute, Stockholm, Sweden; [3]Max Planck Institute Biology of Ageing - Karolinska Institutet Laboratory, Karolinska Institutet, Stockholm, Sweden

**\*For correspondence:**
joanna.rorbach@ki.se (JR);
amunts@scilifelab.se (AA)

[†]These authors contributed equally to this work

**Competing interest:** The authors declare that no competing interests exist.

**Abstract** The mitoribosome regulates cellular energy production, and its dysfunction is associated with aging. Inhibition of the mitoribosome can be caused by off-target binding of antimicrobial drugs and was shown to be coupled with a bilateral decreased visual acuity. Previously, we reported mitochondria-specific protein aspects of the mitoribosome, and in this article we present a 2.4-Å resolution structure of the small subunit in a complex with the anti-tuberculosis drug streptomycin that reveals roles of non-protein components. We found iron–sulfur clusters that are coordinated by different mitoribosomal proteins, nicotinamide adenine dinucleotide (NAD) associated with rRNA insertion, and posttranslational modifications. This is the first evidence of inter-protein coordination of iron–sulfur, and the finding of iron–sulfur clusters and NAD as fundamental building blocks of the mitoribosome directly links to mitochondrial disease and aging. We also report details of streptomycin interactions, suggesting that the mitoribosome-bound streptomycin is likely to be in hydrated gem-diol form and can be subjected to other modifications by the cellular milieu. The presented approach of adding antibiotics to cultured cells can be used to define their native structures in a bound form under more physiological conditions, and since streptomycin is a widely used drug for treatment, the newly resolved features can serve as determinants for targeting.

## Editor's evaluation

This manuscript describes high-resolution structures of the small subunit of the human mitochondrial ribosome that reveal for the first time a number of physiologically important small molecules (GTP and NAD+/NADH) and iron-sulfur clusters integrated into the small subunit architecture. In the two structures, the authors also describe interactions of the small subunit with the antibiotic streptomycin that reveal how this antibiotic may be chemically modified by the cellular environment. The chemical-level detail revealed by these structures lays a foundation for future efforts to understand the basis for mitochondrial function in human health and disease.

## Introduction

The human mitoribosome has a distinct structure, it receives mRNAs with involvement of Leucine-rich PPR-motif-containing protein (LRPPRC) and synthesizes 13 respiratory chain proteins delivered to the inner mitochondrial membrane via the OXA1L insertase (*Itoh et al., 2021*; *Singh et al., 2022*). Cryo-EM has been instrumental in the structure determination of the mitoribosome, however, the description is currently limited to mitoribosomal proteins, rRNA, and associated factors, whereas

important metabolic cofactors of potential therapeutic interest are largely excluded (*Aibara et al., 2020*). In addition, dysfunction of the human mitoribosome can be caused by the off-target binding of antimicrobials to structured rRNA core that is similar to bacteria, which can lead to clinical symptoms of deafness, neuropathy, and myopathy. However, off-target binding studies are currently not incorporated in pharmaceutical companies' research and development pipelines. The antimicrobial binding has also been used to suppress glioblastoma stem cell growth (*Sighel et al., 2021*), suggesting that repurposing of mitoribosome-targeting antibiotics offers a therapeutic option for tumors (*Vendramin et al., 2021*). Thus, the structure-based design of molecules that bind RNA can be used to optimize physicochemical properties and has the potential of improving pharmacokinetics and potency.

The aminoglycoside streptomycin targets the ribosomal small subunit (SSU), but the therapy with streptomycin was also shown to be coupled with a bilateral decreased visual acuity with central scotomas and an altered mitochondrial structure (*Kogachi et al., 2019*). Moreover, patients carrying mitochondrial DNA mutations in the 12S rRNA gene, such as 1555A>G or 1494C>T are more prone to aminoglycoside-induced ototoxicity (*Gao et al., 2017*). To minimize toxic off-target effects, the approaches based on in silico modeling employing high-resolution single-particle cryo-EM structures can be used. Although the sensitivity of mitoribosomes to antimicrobials has been documented, no detailed structural information elucidating specific molecular interactions is available, thus mechanistic details remain unknown.

Therefore, in this study, we set to solve the structure of complex of streptomycin with the mitoribosomal SSU at a resolution level that can detect cofactors and modifications in order to provide new structural information on potential mitoribosomal constituents. We explored fine structural elements of the antibiotic and determined unique features of the mitoribosome, such as iron–sulfur clusters positioned between different proteins, nicotinamide adenine dinucleotide (NAD), and a new guanosine triphosphate (GTP)-binding site.

## Results

### Structure determination

To characterize the binding under close to physiological conditions, we added streptomycin to cultured human embryonic kidney 293T (HEK293T) cells at a final concentration of 100 µg/ml and not to any of the biochemical purification steps. This approach implies that the antimicrobial would have to be imported into mitochondria, and therefore has an advantage over in vitro complex formation (that we performed as a control), as more native inhibitory properties would be preserved. Mitochondria were isolated, and the mitoribosomal SSU was purified in the presence of 5′-guanylyl imidodiphosphate (GMPPNP) and subjected to a cryo-EM analysis. Monosome and large subunit (LSU) particles were removed during 2D classification, and the remaining particles underwent 3D auto-refinement and 3D classification with local angular search with a solvent mask to remove poorly aligned particles. The resolution was further improved by applying contrast transfer function (CTF) refinement including beam-tilt, per-particle defocus, and per-micrograph astigmatism, followed by Bayesian polishing in RELION 3.1 (*Zivanov et al., 2020*). Particles were then separated into multi-optics groups based on acquisition areas and the date of data collection. A second round of CTF refinement (beam-tilt, trefoil, and fourth-order aberrations, magnification anisotropy, per-particle defocus, per-micrograph astigmatism) was performed, followed by 3D auto-refinement. Finally, to improve the local resolution, local-masked 3D auto-refinements were systematically applied (*Figure 1—figure supplement 1*).

The resulting structure of the SSU with bound streptomycin was determined at 2.4 Å resolution (*Figure 1—figure supplement 1* and *Table 1*). This represents a substantial improvement of the X-ray crystal structures at 3.0–3.5 Å resolution of the in vitro formed complexes of streptomycin with *T. thermophilus* ribosome (*Carter et al., 2000*; *Demirci et al., 2013*), as well as the previous cryo-EM structures at ~3 Å of the human mitochondrial SSU (*Khawaja et al., 2020*; *Itoh et al., 2022*). The higher resolution allowed us to detect a possible modification of the bound streptomycin and previously unknown cofactor components of the mitoribosome (*Figure 1*). Although compared to bacteria, human mitoribosomal rRNA is known for its substantial reduction in size, we detected and modelled a specific nucleotide insertion C1048 paralleled by a stabilizing tetraamine spermine (SPM) and NAD. Four other cofactors reported in our structure are: two iron–sulfur clusters (2Fe–2S),

**Table 1.** Cryo-EM collection, processing, model refinement, and validation statistics.

| Data collection and processing | Native SSU:streptomycin complex | In vitro formed SSU:streptomycin complex |
|---|---|---|
| Microscope | Titan Krios | Titan Krios |
| Detector | K2 Summit | K3 Summit |
| Magnification | 165,000 | 105,000 |
| Voltage [kV] | 300 | 300 |
| Total electron exposure [$e^-$/Å$^2$] | 30–32 | 40 |
| Defocus range [μm] | −0.2 to −3.6 | −0.2 to −3.7 |
| Pixel size [Å] | 0.83 | 0.846 |
| Symmetry imposed | $C_1$ | $C_1$ |
| Final particle | 885,199 | 899,952 |
| Resolution [Å] (overall/body/shoulder/platform/back/tail/head-AP/head-PE/mS39/mtIF3) | 2.40/–/2.23/2.28/2.29/2.41/2.26/2.39/2.57/2.20 | 2.31/2.28/–/–/–/–/–/–/–/– |
| Map-sharpening $B$-factor [Å$^2$] (overall/shoulder/platform/back/tail/head-AP/head-PE/mS39/mtIF3) | −53/–/−47/−49/−56/−60/−52/−58/−68/−54 | −50/−49/–/–/–/–/–/–/–/– |
| **Refinement** | | |
| Model composition | | |
| Total atoms (non-hydrogen/hydrogen) | 71,880/59,406 | |
| Chains (RNA/protein) | 1/31 | |
| RNA residues (non-modified/m$^4$C, m$^5$C, m$^5$U, m$^6_2$A) | 950/1/1/1/2 | |
| Protein residues (non-modified/$N$-acetylAla/$O^1$-methylisoAsp) | 5915/2/1 | |
| Metal ions (Mg$^{2+}$/K$^+$/Zn$^{2+}$) | 62/21/1 | |
| Ligands (2Fe–2S/ATP/GMPPNP/NAD/spermine/streptomycin) | 2/1/1/1/1/1 | |
| Waters | 3087 | |
| Model to map CC (CC$_{mask}$/CC$_{box}$/CC$_{peaks}$/CC$_{volume}$) | 0.90/0.79/0.79/0.88 | |
| Resolution [Å] by model-to-map FSC, threshold 0.50 (masked/unmasked) | 2.18/2.18 | |
| Average $B$-factor [Å$^2$] (RNA/protein/metal ion and ligand/water) | 36/48/32/32 | |
| R.m.s. deviations, bond lengths [Å]/bond angles [°] | 0.002/0.408 | |
| **Validation** | | |
| Clash score | 1.39 | |
| Rotamer outliers [%] | 0 | |
| Ramachandran plot [%] (favored/ allowed/disallowed) | 98.01/1.95/0.03 | |
| CaBLAM outliers [%] | 0.71 | |
| $C_\beta$ outliers [%] | 0 | |
| MolProbity score | 0.87 | |
| EMRinger score | 6.16 | |
| PDB/EMDB accession code | 7P2E/EMD-13170 | –/EMD-15542 |

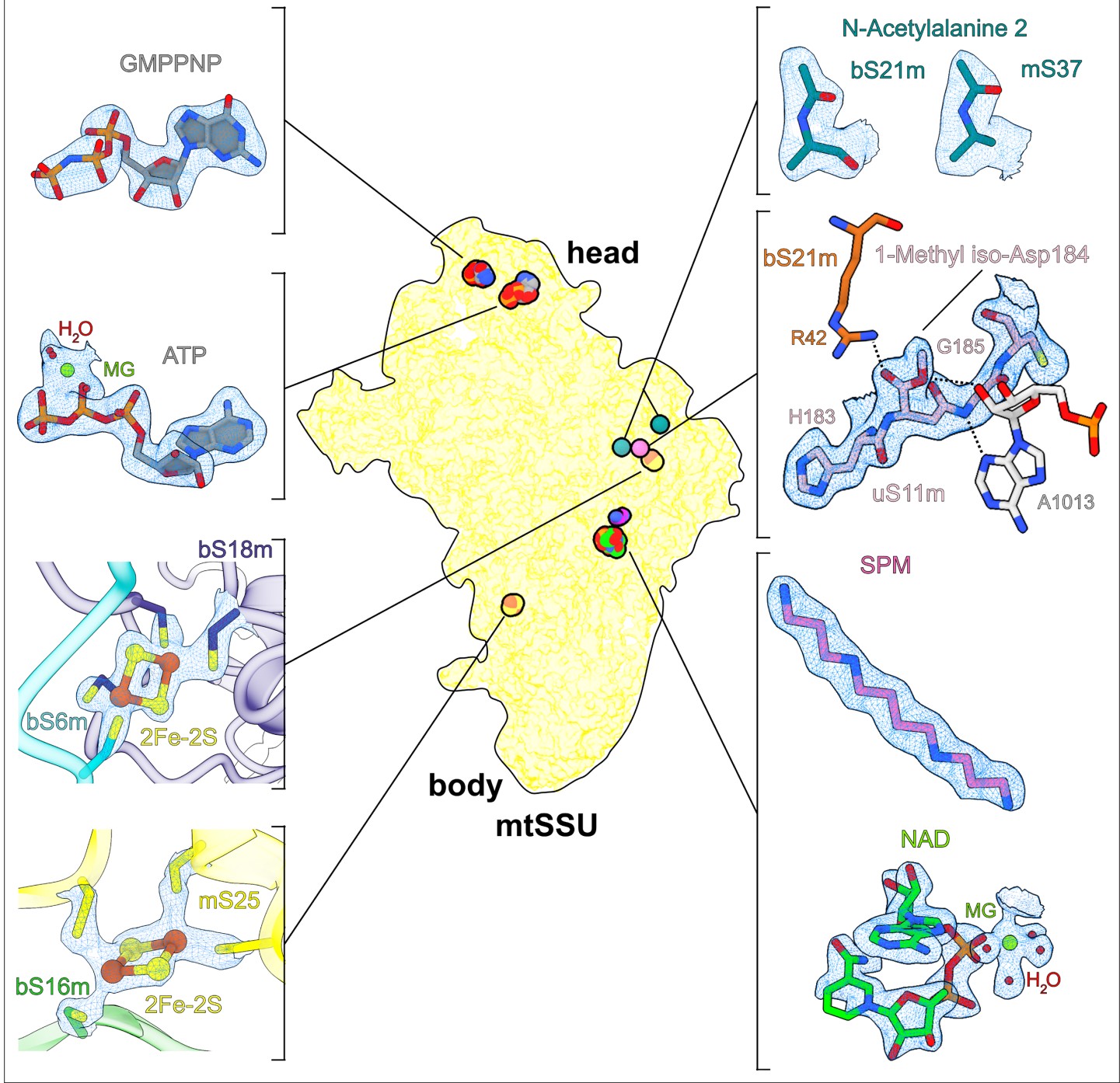

**Figure 1.** Overview of cofactors and modifications of the mitoribosomal small subunit. Outlined view indicates the relative positions of identified cofactors and modifications. Cryo-EM densities, models, and surrounding environment are shown for GMPPNP, adenosine triphosphate (ATP), 2Fe–2S clusters, protein modifications, spermine, and nicotinamide adenine dinucleotide (NAD).

The online version of this article includes the following figure supplement(s) for figure 1:

**Figure supplement 1.** Cryo-EM data collection and processing.

**Figure supplement 2.** Cryo-EM data collection and processing for in vitro reconstituted small subunit (SSU):streptomycin complex.

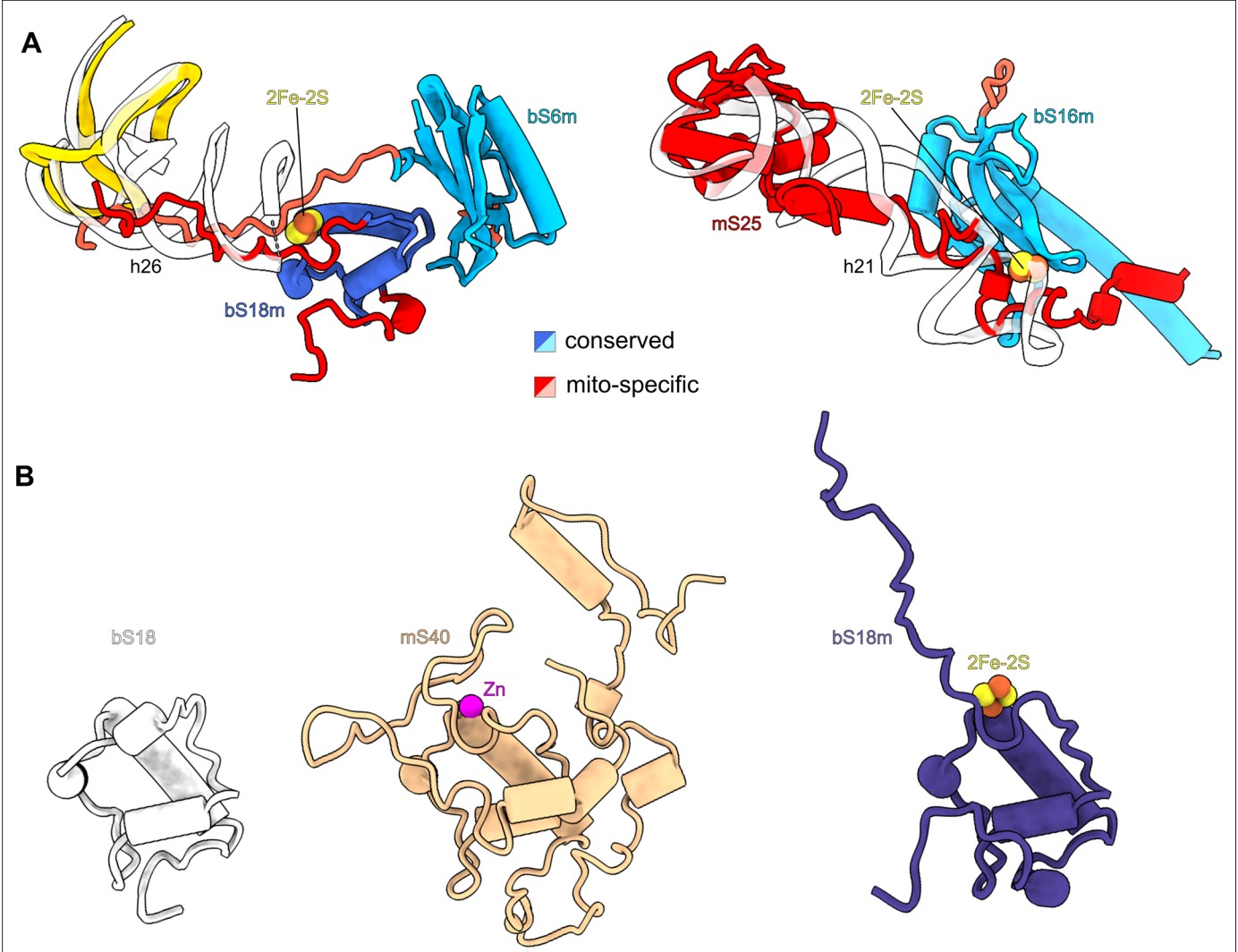

**Figure 2.** Iron–sulfur clusters bridge mitoribosomal proteins. (**A**) Each of the two 2Fe–2S clusters is coordinated by mitochondria-specific components (red) from two different proteins. Superposition of 2Fe–2S cluster-binding sites in the human mitoribosome with corresponding sites in *E. coli* ribosome (PDB ID: 7K00, white) reveals that mitoribosomal protein elements participating in 2Fe–2S coordination compensate for missing rRNA helices. (**B**) The conserved overall fold of bS18 contributes to $Zn^{2+}$ or 2Fe–2S binding in the mitoribosome.

adenosine triphosphate (ATP), and the added GMPPNP. In addition, three protein modifications could be detected from the density.

## Iron–sulfur cluster involved in interactions between mitoribosomal proteins

The structure contains two iron–sulfur clusters, and their coordination involves mitochondria-specific protein elements (*Figure 1*, *Figure 2A*). The positions of both clusters are correlated with rRNA reduction compared to the bacterial counterpart. We identified the clusters in two ordered densities, adjacent to sulfhydryl groups of cysteine residues in four different mitoribosomal proteins (*Figure 1*). Each of the clusters is composed of four atoms organized in a planar square, with two of the atoms bound by two cysteinyl S atoms of the mitoribosomal proteins. The map shows a weaker density in the center of the planar squares. Such a chemical arrangement corresponds to the 2Fe–2S cluster, where iron atoms with bridging sulfur atoms are coordinated by proteins (*Beinert et al., 1997*). A unique feature of our structure is that both 2Fe–2S clusters are coordinated not by the same protein but by two different proteins, which is an unprecedented occurrence in any other known Fe–S systems.

Cluster-1 (chain P, FES 201 in the PDB) links extensions of bS18m and bS6m close to the platform region (*Figures 1 and 2A*). The former forms a mitochondria-specific intersubunit protein–protein contact that contributes to sampling a larger conformational space than the bacterial counterpart. In *T. thermophilus* the binding of bS6:bS18 requires a formation of the central RNA domain first (*Agalarov et al., 2000*), and therefore occurs during an advanced assembly process. Cluster-2 (chain T, FES 201 in the PDB) links the mitochondria-specific protein mS25 with bS16m in the lower body (*Figures 1 and 2A*). Despite being a peripheral mitoribosomal protein, the correct assembly of mS25 is crucial (*Bugiardini et al., 2019*), and mutations in its binding partner bS16m are associated with corpus callosum agenesia, hypotonia, and fatal neonatal lactic acidosis (*Miller et al., 2004*). Interestingly, we found another homolog of bS18m on the SSU: mitochondria-specific protein mS40. However, unlike its counterpart, mS40 binds a zinc ion in our structure and not a 2Fe–2S cluster, and it is not involved in protein bridging (*Figure 2B*).

The comparison with *E. coli* ribosome (PDB ID: 7K00) shows that the 2Fe–2S clusters are found where bacterial rRNA is missing (*Figure 2A*), suggesting that the incorporation of the iron–sulfur clusters provides a mechanism by which protein stabilization is coupled with the structural compensation for the rRNA deletion. Together, our observation of iron–sulfur clusters bridging between different proteins suggests a role in the mitoribosomal assembly, rather than a catalytic function.

## NAD is associated with a single rRNA nucleotide insertion

Another notable feature of the map is the presence of two related densities associated with rRNA that we assigned as spermine and NAD (*Figure 1*). In the SSU-body, we found a spermine buried within the reduced rRNA core, bound to the rRNA h20. Surprisingly, despite the general conception that human mitochondrial rRNA only has large deletions, we found an insertion of the nucleotide C1048 in the mitochondrial genome. In the structure, the addition of this residue disrupts the base-paring that supports the h20 architecture, and the residues A1047 and C1048 are flipped out (*Figure 3A, B*). Spermine is then inserted through h20 and contacts the backbone of C1048 for stabilization. Spermine forms salt bridges with the phosphate groups of U946, U1044, G1045, and C1048 and forms hydrogen bonds with the base moieties of U944 and G945. While polyamines, like spermine, are commonly found in ribosomes (*Watson et al., 2020*; *Zgadzay et al., 2022*), we also detected an NAD at the same site, and both cooperatively compensate for the lack of internal rRNA interactions in this region as a result of the insertion (*Figure 3A, B*). NAD adopts a compact conformation, where its two base rings are stacked on each other. The adenine ring forms a base pair with U948 and is further stacked on A1046. NAD also interacts with A781, A782, A1047 (through Mg-ion coordination), and C1048 of rRNA and Tyr196 of uS15m. The density is suggestive that the ring may be a mixture of oxidized and reduced states, although the structure does not reveal whether it might have a regulatory function on the mitoribosome or affect the local environment.

## New GTP-binding site in the head of the mitoribosomal SSU

The SSU head hosts arguably the most enigmatic mitoribosomal protein mS29 – a putative intrinsic GTPase, and therefore it is speculated that it may play a role in the translation cycle (*Amunts et al., 2015*; *Greber et al., 2015*; *Ott et al., 2016*). To clarify its structure, we performed our studies in the presence of a nonhydrolyzable analog GMPPNP. Careful inspection of the map at the improved resolution revealed that the reported nucleotide pocket binds ATP with an $Mg^{2+}$ ion, and the previously modeled GDP was misassigned due to the limited resolution. Our argument is based on N6-amino and N1-imino groups of the adenine ring that form hydrogen bonds with the backbone carbonyl and NH groups of Met100, respectively (*Figure 4B*). This would be incompatible with guanine, since its O6-carbonyl and N1-amino groups impart repulsive interactions with the backbone carbonyl and NH groups of Met100, respectively (acceptor with acceptor and donor with donor). Therefore, this binding site does not seem to bear a functional GTPase activity, which is also consistent with a fungal mitoribosome (*Itoh et al., 2020*).

Notably, distinct from the reported binding site, we also identified a density that is not continuous with any of the assigned mS29 residues, and it corresponds to the added GMPPNP (*Figure 1*). The density is found in the hydrophobic cavity formed by Tyr173, Tyr208, Trp210, and Ile242, where it stabilizes a beta-hairpin (residues 208–216) on the top of mS29 in the SSU head (*Figure 4B*). The local environment for the base specificity reveals that N1-amino and O6-carbonyl groups form hydrogen

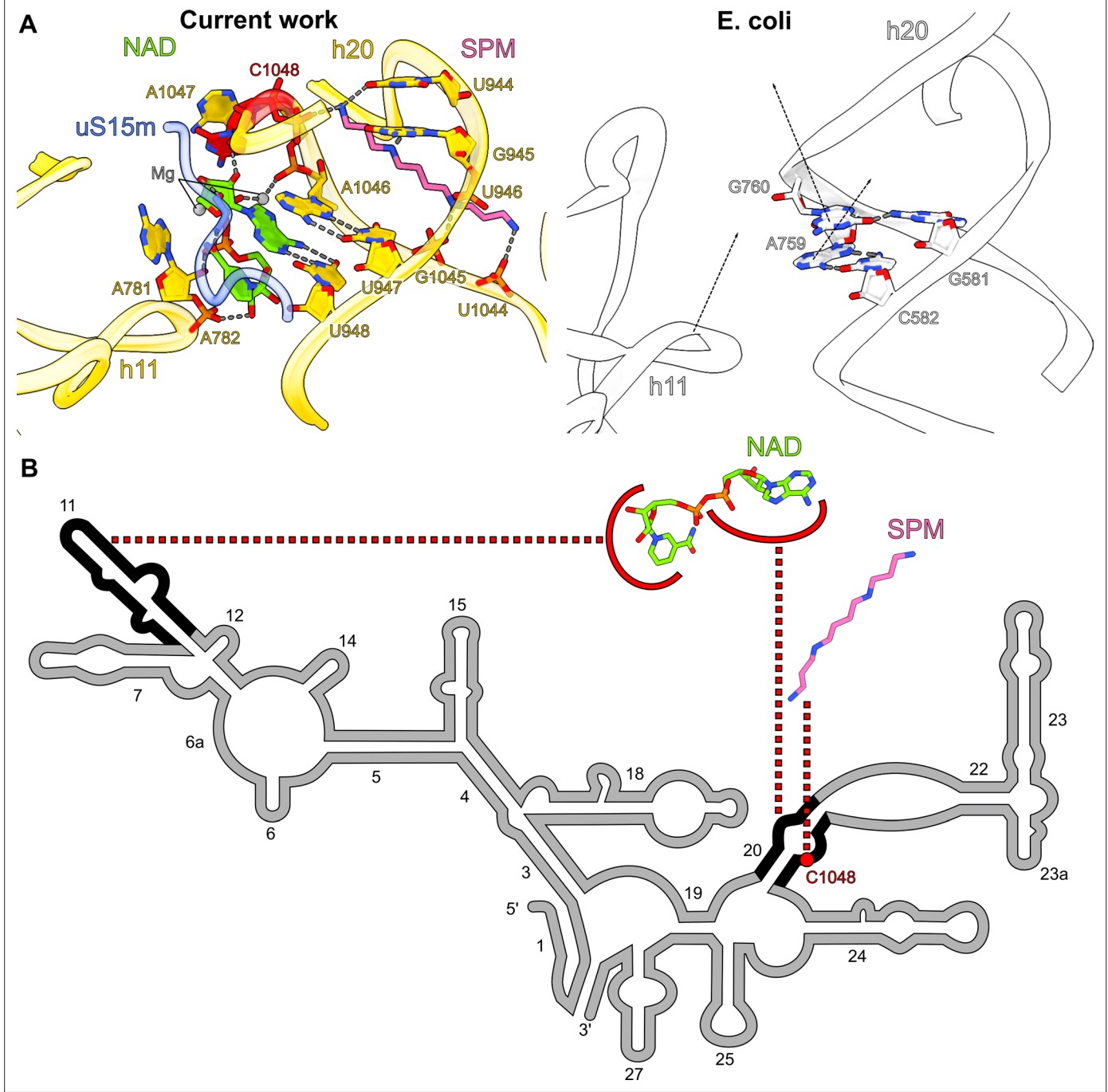

**Figure 3.** Spermine-NAD functionally compensates for rRNA and protein alterations. (**A**) Comparison of the spermine-NAD binding site with *E. coli* ribosome (PDB ID: 7K00, white) shows that it provides structural stability that compensates for rRNA alterations. (**B**) 2D diagram of the SSU head rRNA (grey) with regions stabilized by NAD and spermine indicated (black); the associated insertion C1048 is highlighted in red.

bonds with the backbone NH and carbonyl groups of Val209, respectively, while the N2-amino group forms hydrogen bonds with the backbone carbonyl group of Val209 and the side chain carboxyl group of Asp238. The ribose moiety has hydrogen bonds with Asp238 and Asn292 and the phosphate groups have polar interactions with Tyr173, Arg177, Lys245, His291, and Lys295. Thus, the chemical environment and the density features suggest a separate GTP/GDP-binding pocket. The network of interactions indicates that GTP would be favored over GDP in the binding pocket, and this explains why the SSU has a high affinity for GTP (*Denslow et al., 1991*).

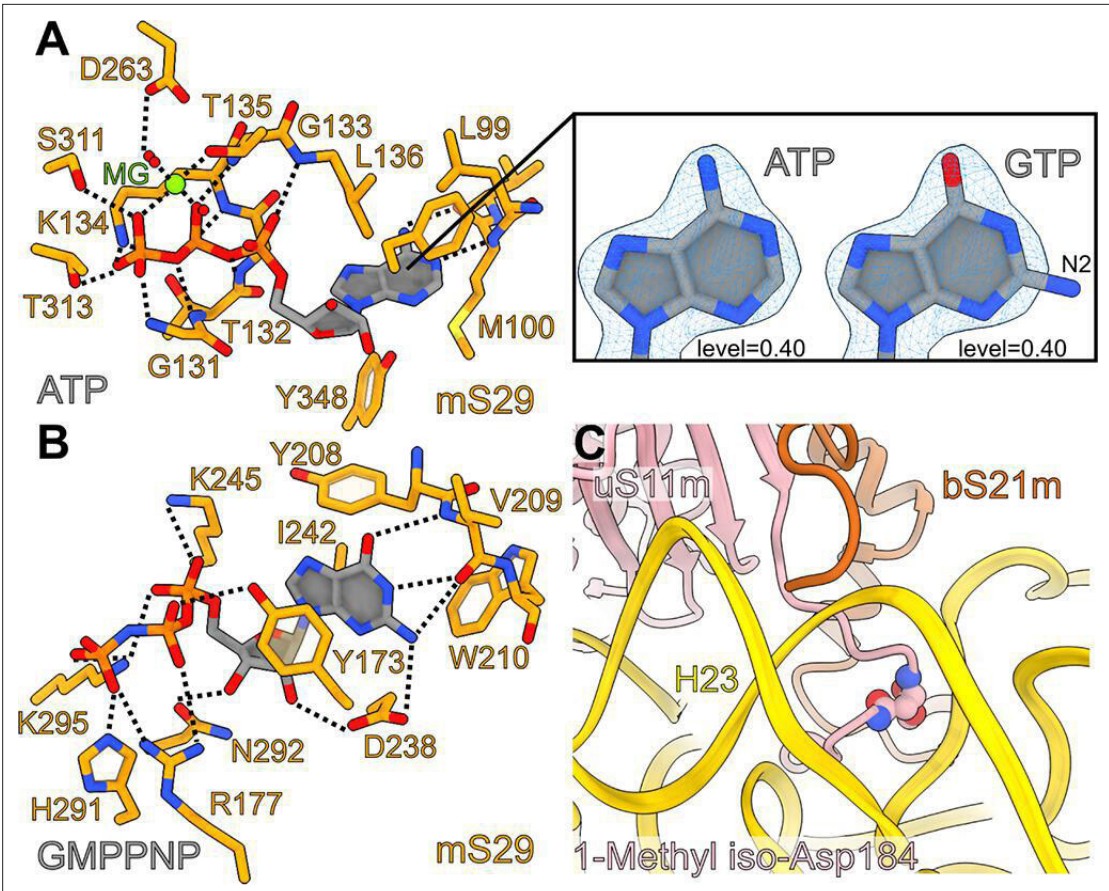

**Figure 4.** Interactions of adenosine triphosphate (ATP), GMPPNP in mS29, and 1-methyl iso-aspartate in uS11m. (**A**) Interactions of the bound ATP with the surrounding residues of mS29. The right panel shows the fitting of adenine versus guanine to the density, featuring the amine group N2 on carbon 6 that does not fit when guanosine triphosphate (GTP) is modeled. (**B**) The newly identified binding pocket and the interactions of GMPPNP with the surrounding residues of mS29. (**C**) Bending of uS11m loop at the site of occurrence of 1-methyl iso-Asp 184 of uS11m to facilitate interactions with helix h23.

## Protein modifications identified from the density

Some posttranslational chemical modifications of ribosomal proteins can be identified directly from the density (*Watson et al., 2020*; *Cottilli et al., 2022*), and our study reveals a mainchain modification 1-methyl-isoaspartate at position 184 in uS11m, and the N-terminal acetylations in bS21m and mS37 (*Figure 1*). The density around Asn184 was incompatible with the placement of an asparagine residue as indicated by the primary sequence. Instead, an isoaspartate (iso-Asp) was modeled, so that the β-carbonyl group forms backbone peptide linkage with Gly185, and the α-carboxyl group is exposed, in agreement with the modification detected in *E. coli* (*David et al., 1999*; *Watson et al., 2020*) and plants (*Cottilli et al., 2022*). Due to the elongated backbone peptide by one methylene group at the iso-Asp, the loop is kinked between His183 and iso-Asp184. This specific kinked loop interacts with rRNA h23–24, and thereby likely contributes to its stability (*Figure 4C*). Furthermore, an additional density was observed continuous with an α-carboxyl oxygen atom, and thus it corresponds to a non-hydrogen atom, which led us to replace the isoaspartate with 1-methyl-isoaspartate that is known as a product of the enzyme L-isoaspartyl *O*-methyltransferase (*Reissner and Aswad, 2003*). The backbone peptide and the α-methoxy group form hydrogen bonds with the adenine and ribose moieties of A1013 rRNA, respectively, while the α-carbonyl group forms a hydrogen bond with bS21m Arg42 (*Figure 1*).

For two mitoribosomal proteins bS21m and mS37, to model *N*-acetylations, we completed the model up to the corresponding N-terminal residues, and the remaining densities were too small to accommodate a methionine, but consistent in size with an acetyl group that is represented by three

non-hydrogen atoms. *N*-Acetylated residues were then added and adjusted manually taking into consideration the immediate chemical environment. Thus, in bS21m and mS37, the first methionine is removed and the second residue is *N*-acetylated, which is a typical product of the N-terminal acetyl-transferases (*Starheim et al., 2009*).

## Streptomycin binding with a possible modification

We could model streptomycin unambiguously into the map and identify the interactions and solvation that stabilize it in the binding pocket (*Figure 5A-C*, *Figure 5—figure supplement 1*). The chemical structure of streptomycin is comprised of three components linked by ether bonds: streptidine (scyllo-inositol with two hydroxyl groups substituted by guanidino groups), streptose (3-formyl-4-methyl tetrose), and *N*-methyl-L-glucosamine (*Figure 5A, B*). However, streptomycin can also be subjected to modifications under different environmental conditions that might affect its chemical properties in the cellular milieu (*Abraham et al., 1946*; *Stern et al., 2018*; *Alekseeva et al., 2019*), and thus be of a potential therapeutic interest.

In our structure, the density for the streptose moiety reveals a series of unexpected features (*Figure 5C*): (1) it is generally the poorest resolved component of streptomycin, (2) particularly, the methyl group of streptose is not well covered by the density, (3) the previously modeled aldehyde group (*Carter et al., 2000*; *Demirci et al., 2013*) appears as a loosely bound density, and (4) the hydroxyl group is not well resolved. To further clarify the model, we calculated atomic *B*-factors estimated by reciprocal space refinement, which supports the idea that the streptose moiety is relatively flexible (*Figure 5C*). The density surrounding the methyl group also suggests the stereochemistry for the methyl group might have been inverted, possibly by enzymatic activity within cells, however, the map quality is not unambiguous in this region (*Figure 5D*). The density replacing the aldehyde group is within the hydrogen bonding distance of four phosphate groups of rRNA (C898, G899, A1166, and A1167). Given that aldehyde has no hydrogen to provide for H-bonding phosphates, the ribosome-bound streptomycin is likely to be in the hydrated gem-diol form rather than in the free aldehyde form.

This hydration has been previously reported by an NMR study of the free unbound state in an aqueous solution (*Blundell et al., 2013*). Additionally, previously reported sub-2 Å resolution X-ray electron density maps of the streptomycin bound to adenylyltransferase AadA (PDB: 5LUH, *Stern et al., 2018*) and an aminoglycoside phosphotransferase APH(3″)-Id (PDB: 6FUX, *Alekseeva et al., 2019*) show clear branched densities and interactions of the aldehyde moiety (*Figure 5—figure supplement 2*). This potentially also indicates a gem-diol, although they are interpreted as two alternative conformations of the aldehyde group in the published models (*Stern et al., 2018*; *Alekseeva et al., 2019*). On the other hand, the density of the streptose moiety in our map is unclear and therefore we cannot exclude the possibility of unidentified modification of streptomycin.

## Comparison between native and in vitro bound streptomycin

To find out whether the streptomycin streptose moiety alteration originated in its chemical production or was affected by the cellular milieu, we next purified the SSU from untreated cells and added streptomycin in molar excess in vitro to determine its bound structure. In contrast to our previous experiment, this approach implies that the antimicrobial would maintain its original properties without being subjected to the cellular environment. After 30 min of incubation, the sample was subjected to cryo-EM analysis, and the resulting structure of the SSU with in vitro bound streptomycin was determined at 2.3-Å nominal resolution (*Figure 1—figure supplement 2* and *Table 1*). Unlike in the native structure, the streptose moiety is well resolved, and no chirality flip is observed around the methyl group (*Figure 5D*). The density around the aldehyde moiety is branched and supports the modeling of one stabilized rotamer with the oxygen atoms within hydrogen bond distance from backbone phosphates of rRNA residues G898, G899, A1166, and A1167 and an adjacent water molecule (*Figure 5D*). This indicates that the aldehyde is in the hydrated gem-diol state and participates in the potential interactions. Overall, no modification except for the aldehyde hydration is present in the streptomycin added in vitro to purified SSU.

To further confirm the chemical structure of the streptomycin modeled here, we next performed mass spectrometry analysis using a dual jet stream electrospray ion source operating in positive ion mode (*Figure 5—figure supplement 3*). The mass to charge ratio (*m/z*) is consistent with the cryo-EM

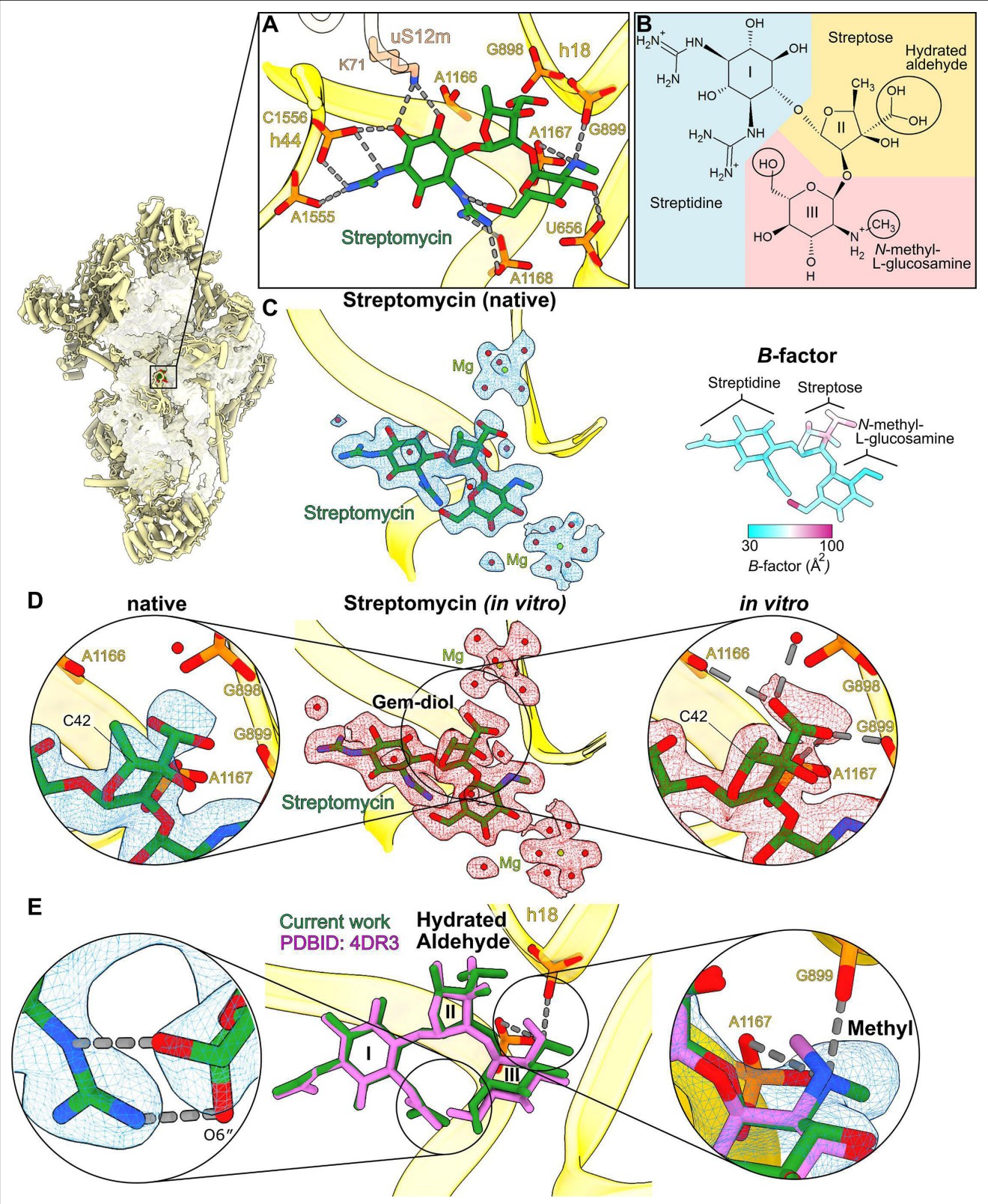

**Figure 5.** High-resolution features of streptomycin binding to the mitoribosome. (**A**) Streptomycin interacts with uS12m and backbone phosphates of helices h18 and h44. (**B**) Chemical structure of the hydrated gem-diol form of streptomycin. (**C**) Left, density and model of the natively bound streptomycin along with surrounding water molecules and Mg²⁺ ions. Right, atomic B-factor distribution of the bound streptomycin shows higher relative flexibility of the streptose moiety. (**D**) Middle, density (red) and model of the in vitro formed complex. Left, zoom-in of the native state shows

*Figure 5 continued on next page*

*Figure 5 continued*

that the density of the gem-diol moiety is not well resolved. Right, zoom-in of the in vitro complex shows that the streptose moiety is better resolved with a defined orientation of the gem-diol moiety. A second notable difference is the inversion of chirality of C42, where the methyl group of streptose is located. (**E**) Comparison with the previously reported structure of *T. thermophilus* ribosome (PDB ID: 4DR3, pink). Close-ups of the chemical interactions of O6″ *N*-methyl-L-glucosamine in its alternative conformations (left zoom-in); and methyl moieties (right zoom-in) with their densities resolve discrepancies of the methyl-group orientation in the previous studies.

The online version of this article includes the following figure supplement(s) for figure 5:

**Figure supplement 1.** Densities of streptomycin interactions with the small subunit (SSU).

**Figure supplement 2.** Comparison of the mitoribosome-bound streptomycin with the published X-ray electron density maps.

**Figure supplement 3.** High-resolution mass spectrometry (HRMS) analysis for streptomycin.

**Figure supplement 4.** Alternative states of h44 rRNA. The density suggests that the residues colored pink in h44 adopt two alternative states (left and right).

analysis of in vitro bound streptomycin displaying the highest peak at ~600.28, which corresponds to the hydrated gem-diol form. However, another less-abundant peak is found at *m/z* value of ~582.2, which corresponds to the exact *m/z* of streptomycin. This suggests that both forms of soluble streptomycin, hydrated and non-hydrated are retained in solution (*Figure 5—figure supplement 3*). Interestingly, in the biosynthesis process of streptomycin by the bacterium *Streptomyces griseus*, the dehydrogenation leading to the aldehyde formation is the last of 27 assembly steps, followed by the compound release from the bacteria for activation by StrA (*Flatt and Mahmud, 2007*). A putative gene product that would mediate this transition is unknown.

## Structural basis for streptomycin toxicity

Comparison with the previous model of streptomycin-bound *T. thermophilus* ribosome (*Demirci et al., 2013*) further shows that the methyl group and the O6″ hydroxyl group have different conformations on the *N*-methyl-L-glucosamine moiety (*Figure 5E*, left panel). Our high-resolution structure indicates the chemically more favorable conformation with the amino group at the 2″ position (protonated secondary amine), forming two hydrogen bonds/salt bridges with two backbone phosphates of rRNA (*Figure 5E*, right panel). We also found two water-coordinating magnesium ions (*Figure 5C*).

With respect to the mutations 1494C>T and 1555A>G in h44, the human mitochondrial rRNA appears to be more flexible in this region due to a loss of the G:C base pair at the 1494:1555 position. However, the two mutations introduce Watson–Crick base pairs that limit the rRNA local flexibility and resemble the corresponding binding site in bacteria that is implicated in aminoglycoside hypersensitivity (*Ippolito et al., 2008*; *Wilson et al., 2008*). Interestingly, the density indicates two alternative states in this region (residues 1491–1495 and 1558–1559), particularly A1558 is flipped out in one state, whereas its base is stacked between G1559 and A1492 in the other state, causing a shift of the residues in the other strand (*Figure 5—figure supplement 4*). Since this feature indicates a flexibility that might be related to recruitment of mRNA for translation (*Khawaja et al., 2020*), a possible mechanistic insight from our work is that by directly interacting with A1555 and C1556 on the mitoribosome, the streptomycin binding decreases that flexibility, which further contributes to translation inhibition.

## Discussion

In this work, we present the cryo-EM reconstructions of the native and in vitro bound complexes of the mitoribosomal SSU with streptomycin at 2.4 and 2.3 Å resolution, respectively. We discovered distinct structural elements of the mitoribosome, including 2Fe–2S clusters and NAD. The comparison with bacteria allowed us to propose that 2Fe–2S clusters play a structural role, being organized between mitoribosomal proteins to stimulate a tighter association. This phenomenon likely reflects the importance of mechanical properties, such as the stability of newly added proteins in the regions where rRNA has been deleted during evolution. Therefore, the clusters facilitate protein complex formation to support the assembly of the human mitoribosome. Although there is evidence that iron–sulfur clusters contribute to the structural stability of the apoproteins into which they are integrated, the involvement in maintaining a complex of proteins was not reported before. It is unknown whether

the facilitation of inter-protein interactions that we found is unique to the mitoribosome or represents a more general characteristic. One of the implications of our finding is that since iron–sulfur clusters are not stable in an aqueous solution without being bound to a protein, there might be yet to be characterized chaperones required for the assembly on the mitoribosome. Notably, the biogenesis of iron–sulfur complexes is associated with acyl carrier protein and fatty-acid synthesis (*Van Vranken et al., 2016*; *Nowinski et al., 2020*), which we also previously identified as a component of the LSU assembly (*Brown et al., 2017*). Therefore, our data provide a potential regulatory link between the three main metabolic pathways in mitochondria: mitoribosomal biogenesis, iron–sulfur assembly and fatty-acid synthesis. In addition, the mitochondrial disease Friedreich's ataxia that is caused by a depletion of an Fe–S cluster biogenesis factor, leads to deficit in mitochondrial oxidative phosphorylation in patient samples (*Marmolino, 2011*), which might also involve mitoribosomes.

The improved resolution also allowed us to identify and model an NAD, which is a metabolite the depletion of which has been proposed to promote aging and mitochondrial myopathy (*Pirinen et al., 2020*). Apart from acting as an electron carrier in metabolic pathways, NAD is a co-substrate of several regulatory pathways, and its loss induces a pseudohypoxic state (*Gomes et al., 2013*). Here, we show that its presence on the mitoribosome contributes to the rRNA stability, and thus might be important for mitochondria function. Among other newly found cofactors, we re-assign the previously modeled GDP to ATP with an $Mg^{2+}$ ion in the mitoribosomal protein mS29. At the same time, based on the local chemical environment, we identified a new GTP/GDP-binding pocket. This provides essential structural information and an accurate model that in the future will help to further our understanding of GTP binding to the SSU and its consequences for the activity of the human mitoribosome.

Our structures detect specific water molecules and metal ions involved in the streptomycin coordination that exert its effects. The comparison between the two methods of adding the antibiotic (cultured cells vs. purified mitoribosome) allowed us to directly observe small differences in terms of hydration and a possible chirality inversion by the cellular milieu. The notion that hydrated aldehyde is present is supported by mass spectrometry analysis. These observations complement previous models of streptomycin-bound bacterial ribosomes and allow us to propose a refined model of binding that also includes alternative conformations. In addition, two alternative states of rRNA residues are found in the vicinity of the streptomycin site in the mRNA-binding region that is mechanistically important for the translation mechanism. Since our structures suggest those residues would lose their flexibility upon streptomycin association, this can explain the potential toxicity. Given that streptomycin is a drug that is used for the treatment of tuberculosis, these structural studies are informative for designing less-toxic drugs.

Overall, these findings demonstrate that high-resolution cryo-EM of native assemblies with therapeutic compounds combined with a mass spectrometry approach is a powerful method to address potential modifications and identify specifically bound cofactors. Conceptually, the approach of adding drugs to cultured cells to study their binding could be applied to other medically relevant subjects and antibacterial compounds. Here, we used streptomycin to study its off-target binding, and since the sensitivity of the mitoribosome can also be exploited to suppress leukemia (*Skrtić et al., 2011*) and glioblastoma stem cell growth (*Sighel et al., 2021*), advances in structure-based design can lead to specific RNA-targeted small molecules for cancer research as well. The presence of cofactors can also affect the activity or selectivity of a compound, thus more data will be needed to define mitoribosomal profiles in different conditions, especially in the context of mRNA and tRNAs. In addition, since cofactors such as polyamines are implicated in aging and regulation of mitochondrial metabolism, determining their roles on the LSU and under native, biologically relevant conditions will allow to better understand how the functions of mitoribosome might be modulated. The disease phenotypes associated with mitoribosomes still remain poorly understood, yet high-resolution cryo-EM of native samples underscores the importance of modifications and cofactors, and thus defining their structural principles will help to better understand the related pathological dysfunctions.

## Materials and methods
### Sample preparation for native SSU–streptomycin complex

Flp-In T-Rex human embryonic kidney 293T (HEK293T) cell line (Invitrogen) was cultured as described previously (*Khawaja et al., 2020*). The cells have doxycycline-inducible expression of the C-terminally

FLAG-tagged human mitochondrial IF3 with a HRV 3C cleavage cite. The cells were grown in DMEM media (Gibco) containing 10% tetracycline-free Fetal Bovine Serum (FBS), 100 µg/ml uridine, 5 µg/ml blasticidin S, and 100 µg/ml hygromycin B. Penicillin–streptomycin solution at pH 6.2 and osmolality 322 (Thermo Fisher, 15140122) was supplemented to the media with the final concentration of 100 units/ml of penicillin and 100 µg/ml streptomycin. Doxycycline (Sigma-Aldrich) was added with the final concentration of 50 ng/ml to the culture to induce the FLAG-tagged IF3 expression 48 hr prior to the harvest.

Cells were collected, resuspended in an ice-cold hypotonic buffer 0.6 M mannitol, 100 mM Tris–HCl pH 7.5, 10 mM Ethylenediamine tetraacetic acid (EDTA), 0.05% bovine serum albumin (BSA), and ruptured by a dounce homogenizer. The lysate was clarified by centrifugation at 800 × $g$ and 4°C for 10 min and the mitochondria were pelleted from the supernatant by further centrifugation at 8000 × $g$ and 4°C for 15 min. The crude mitochondria were loaded onto the sucrose step-gradient (1.0 M and 1.5 M sucrose, 20 mM Tris–HCl pH 7.5, 1 mM EDTA) and centrifuged for 1 hr at 77,000 × $g$ (25,000 rpm) in a SW41 Ti rotor (Beckman Coulter). The band formed by the mitochondria between 1 and 1.5 M sucrose was collected and resuspended in 10 mM Tris–HCl pH 7.5 in 1:1 ratio. After centrifugation at 8000 × $g$ and 4°C for 15 min, the purified mitochondrial pellet was resuspended in mitochondrial freezing buffer (200 mM trehalose, 10 mM Tris–HCl pH 7.5, 10 mM KCl, 0.1% BSA, 1 mM EDTA), flash-frozen and stored at −80°C (*Aibara et al., 2018*).

The purified mitochondria were lysed by incubating at 4°C for 20 min in the lysis buffer 25 mM HEPES–KOH pH 7.5, 5.0 mM Mg(OAc)$_2$, 100 mM KCl, 2% (vol/vol) Triton X-100, 0.2 mM Dithiothreitol(DTT), 1× cOmplete EDTA-free protease inhibitor cocktail (Roche), 40 U/µl RNase inhibitor (Invitrogen). The lysate was centrifuged at 5000 × $g$ for 5 min at 4°C and the supernatant was added to ANTI-FLAG M2 Affinity Gel (Sigma-Aldrich), equilibrated with the wash buffer (25 mM HEPES-KOH pH 7.5, 5.0 mM Mg(OAc)$_2$, 100 mM KCl, 0.05% N-dodecyl-beta-D-maltoside [β-DDM]). After 3 hr incubation at 4°C, the gel was washed with the wash buffer and the IF3-bound ribosome was eluted by additional incubation of 2 hr with the PreScission protease (GE Healthcare) (2 U/µl). The IF2-GMPPNP mix was added (final concentration of 1 µM IF2 and 0.25 mM GMPPNP) to the IF3-bound SSU eluate ($A_{260}$ = 7.9) together with the fMet-tRNA$^{Met}_i$ (1 µM) and MTCO2 mRNA (1 µM) and incubated for another 30 min at room temperature, as described previously (*Khawaja et al., 2020*).

## Sample preparation of in vitro formed SSU–streptomycin complex

HEK293-derived cells were cultured in Freestyle 293 Expression Medium (Thermo Fisher) in a vented flask shaking at 120 rpm at 37°C under 5% CO$_2$. The culture was scaled up by splitting at a cell density of 3.0 × 10$^6$ cells/ml up to 2 l final volume of the cell culture. The cells were harvested at a density of 3.7–4.0 × 10$^6$ cells/ml by centrifugation at 1000 × $g$ for 10 min and washed with cold phosphate-buffered saline. Next, mitochondria were purified from the collected cells as described above. Purified mitochondria were lysed in the buffer containing 25 mM HEPES–KOH pH 7.45, 50 mM KCl, 20 mM Mg(OAc)$_2$, 2% Triton X-100, 2 mM DTT, supplemented with cOmplete protease inhibitors and RNase inhibitors, and incubated for 20 min at 4°C. The mitochondrial lysate was centrifuged at 20,000 × $g$ for 5 min at 4°C, and subsequently overlayed on top a 10–30% sucrose gradient in the ribosome buffer (25 mM HEPES/KOH pH 7.5, 50 mM KCl, 20 mM Mg(OAc)$_2$, 2 mM DTT). After centrifugation for 15 hr at 79,000 × $g$ in a SW41 Ti rotor (Beckman Coulter), the gradients were fractionated with a Biocomp Fractionator. Fractions corresponding to the monosomes were pooled and concentrated by pelleting at 135,520 × $g$ (55,000 rpm) for 16 hr at 4°C using a TLA55 rotor (Beckman Coulter). The monosome pellet was resuspended in dissociation buffer (50 mM HEPES/KOH pH 7.6, 300 mM KCl, 5 mM Mg(OAc)$_2$, 2 mM DTT) and incubated for 2 hr at 4°C. The suspension was loaded on top of a 10–30% sucrose gradient prepared in dissociation buffer and centrifuged at 54,455 × $g$ for 21 hr in a SW41 Ti rotor. The gradient was subsequently fractionated using a Biocomp Fractionator. The peak corresponding to the SSU was pooled and concentrated using the centrifugal concentrator Vivaspin MWCO 30 000 PES (Sartorius). The purified SSU (150 nM) was incubated with streptomycin solution (~600 µM) (Thermo Fisher, 15140122) for 30 min at room temperature to form the complex.

## Cryo-EM data collection and processing

For the native SSU–streptomycin complex, 3 µl of ~120 nM mitoribosome was applied onto a glow-discharged (20 mA for 30 s) holey carbon grid (Quantifoil R2/2, copper, mesh 300) coated with

continuous carbon (of ~3 nm thickness) and incubated for 30 s in a controlled environment of 100% humidity and 4°C. The grids were blotted for 3 s, followed by plunge-freezing in liquid ethane, using a Vitrobot MKIV (Thermo Fisher). Datasets were collected on a Titan Krios transmission electron microscope operated at 300 keV, using C2 aperture of 70 μm and a slit width of 20 eV on a GIF quantum energy filter (Gatan). A K2 Summit detector (Gatan) was used at a pixel size of 0.83 Å (magnification of ×165,000) with a dose of 29–32 electrons/Å$^2$ fractionated over 20 frames. A defocus range of −0.5 to −3.6 μm was used. More detailed parameters are listed in *Table 1*.

For in vitro formed SSU–streptomycin complex, 3 μl of 150 nM SSU (incubated with streptomycin as described above) was applied onto a Quantifoil R2/2 holey carbon grid coated with continuous carbon (~3 nm thickness), glow-discharged at 20 mA for 30 s. After application of sample, the grid was incubated for 30 s in a controlled environment of 100% humidity at 4°C using a Vitrobot MKIV (Thermo Fisher), blotted for 3 s and plunge-frozen in liquified ethane. Dataset was collected on Titan Krios transmission electron microscope operated at 300 keV, using C2 aperture of 50 μm and a slit width of 20 eV on a GIF quantum energy filter (Gatan). A K3 Summit detector (Gatan) was used at a pixel size of 0.846 Å (magnification of ×105,000) with a dose of 40 electrons/Å$^2$ fractionated over 40 frames. A defocus range of −0.5 to −2.0 μm was used (*Table 1*).

For the processing of cryo-EM data from the native SSU–streptomycin sample, movie frames were aligned and averaged by global and local motion corrections by RELION 3.0 (*Zivanov et al., 2018*). CTF parameters were estimated by Gctf (*Zhang and Gctf, 2016*). Particles were picked by Gautomatch (http://www.mrc-lmb.cam.ac.uk/kzhang). The picked particles were subjected to 2D classification to discard contaminants as well as the LSU and monosome particles. The remaining particles underwent 3D auto-refinement with RELION 3.0 using EMD-10021 as a 3D reference, followed by 3D classification with local angular search with a solvent mask to remove poorly aligned particles. Focused 3D classification with signal subtraction was performed to pool the SSU particles with IF3 but without IF2, as described previously (*Khawaja et al., 2020*). The particles were subjected to 3D refinement and CTF refinement (beam-tilt, per-particle defocus, per-micrograph astigmatism) by RELION 3.1 (*Zivanov et al., 2020*), followed by Bayesian polishing. Particles were then separated into multi-optics groups based on acquisition areas and the date of data collection. The second round of CTF refinement (beam-tilt, trefoil, and fourth-order aberrations, magnification anisotropy, per-particle defocus, per-micrograph astigmatism) was performed, followed by 3D auto-refinement. To improve the local resolution, local-masked 3D auto-refinements were performed (*Figure 1—figure supplement 1*).

For the processing of cryo-EM data from in vitro formed SSU–streptomycin sample, movies were motion corrected using Relion 3.1.1. CTF parameters were estimated using Gctf (*Zhang and Gctf, 2016*). Bad micrographs were removed by manual inspection. Particles were picked in RELION 3.1.1 using 2D class averages as references. Reference-free 2D classification was carried out and particles corresponding to SSU classes were pooled for further processing. Particles were exported to cryo-SPARC v.3 (*Punjani et al., 2017*) and subjected to homogeneous refinement. Aligned particles were exported back into RELION 3.1.1 for 3D classification to sort out a final clean set of SSU particles. SSU particles were then subjected to CTF refinement (beam-tilt, per-particle defocus, per-micrograph astigmatism) followed by Bayesian polishing in RELION 3.1.1. Particles were then separated into multi-optics groups based on acquisition areas followed by a second round of CTF refinement (beam-tilt, per-particle defocus, per-particle astigmatism and fourth-order aberrations, magnification anisotropy). The SSU-body was masked refined to achieve improved local resolutions (*Figure 1—figure supplement 2*).

Reported resolutions are based on applying the 0.143 criterion on the Fourier shell correlation between reconstructed half-maps. Finally, the maps were subjected to *B*-factor sharpening and local-resolution filtering by RELION 3.1, superposed to the overall map and combined for model refinement.

## Model building and refinement

For the SSU bound with streptomycin, the starting model was PDB ID: 6RW4. The manual revision was done using *Coot* 0.8 (*Emsley et al., 2010*). The streptidine (inositol with two hydroxyl groups substituted by guanidino groups) and the *N*-methyl-L-glucosamine parts agreed with the density, while the aldehyde group of streptose part disagrees. Therefore, the hydrated aldehyde was placed based on the reported hydration information (*Blundell et al., 2013*). The density indicates an inversion in the

chirality at C42 (*Figure 5D*) the original chirality is retained in the final model. Alternative conformations supported by the density for the glucosamine moiety of the streptomycin, as well as RNA and protein residues were introduced. The introduced alternative conformations are chain-A (12S rRNA) G902, C1491-C1495, A1558-G1559; chain-B (uS2m) Met175, Ile232; chain-E (bS6m) Arg45; chain-Q (bS21m) Arg50; chain-S (mS23) Ala2-Ser4; chain-T (mS25) Arg160; chain-X (mS29) Lys298; chain-Y (mS31) Gln295. Water molecules were automatically picked by *Coot*, followed by manual revision. Geometrical restraints of modified residues and ligands were calculated by Grade Web Server (http://grade.globalphasing.org) or obtained from the CCP4 library (*Lebedev et al., 2012*). Hydrogens were added to the models except for water molecules by REFMAC5 (*Murshudov et al., 2011*) using the prepared geometrical restraint files. The model was then refined against the composite map using Phenix.real_space_refine v1.18 (*Liebschner et al., 2019*) with global energy minimization with reference restraints (only for non-modified protein residues, using the input model as the reference, sigma 5) and rotamer restraints, without Ramachandran restraints. Validation was done by MolProbity (*Williams et al., 2018*). The statistics are listed in *Table 1*.

## High-resolution mass spectrometry analysis for streptomycin

Before high-resolution mass spectrometry analysis the streptomycin standard stock was diluted to 10 ng/µl in mQ water.

The chromatographic separation was performed on an Agilent 1290 Infinity UHPLC-system (Agilent Technologies, Waldbronn, Germany). 1 µl of the diluted standard was injected onto an Acquity UPLC HSS T3, 2.1 × 50 mm, 1.8 µm C18 column in combination with a 2.1 × 5 mm, 1.8 µm VanGuard precolumn (Waters Corporation, Milford, MA, USA) held at 40°C. The gradient elution buffers were A ($H_2O$, 0.1% formic acid) and B (75/25 acetonitrile:2-propanol, 0.1% formic acid), and the flow rate was 0.5 ml min$^{-1}$. The compounds were eluted with a linear gradient consisting of 0.1–10% B over 2 min, B was increased to 99% over 5 min and held at 99% for 2 min; B was decreased to 0.1% for 0.3 min and the flow rate was increased to 0.8 ml min$^{-1}$ for 0.5 min; these conditions were held for 0.9 min, after which the flow rate was reduced to 0.5 ml min$^{-1}$ for 0.1 min before the next injection.

The compounds were detected with an Agilent 6546 Q-TOF mass spectrometer equipped with a dual jet stream electrospray ion source operating in positive ion mode. Purine (4 µM) and HP-0921 (Hexakis(1*H*,1*H*,3*H*-tetrafluoropropoxy)phosphazine) (1 µM) were infused directly into the MS at a flow rate of 0.05 ml min$^{-1}$ for internal mass calibration and accurate mass measurements, the monitored ions were purine *m/z* 121.05; HP-0921 *m/z* 922.0098. The gas temperature was set to 150°C, the drying gas flow to 8 l min$^{-1}$ and the nebulizer pressure 35 psig. The sheath gas temp was set to 350°C and the sheath gas flow 11 l min$^{-1}$. The capillary voltage was set to 4000 V in positive ion mode. The nozzle voltage was 300 V. The fragmentor voltage was 120 V, the skimmer 65 V, and the OCT 1 RF Vpp 750 V. The collision energy was set to 0 V. The *m/z* range was 70–1700, and data were collected in centroid mode with an acquisition rate of 4 scans s$^{-1}$ (1977 transients/spectrum). A second injection of the standard was performed to achieve MSMS fragmentation spectra by auto MSMS at collision energies 10, 20, and 40 V.

## Accession codes

The cryo-EM density maps and atomic coordinates for the native and in vitro SSU–streptomycin complex have been deposited in the Electron Microscopy Data Bank (EMDB) and Protein Data Bank (PDB) under accession codes EMD-13170, EMD-15542, and 7P2E.

## Acknowledgements

Cryo-EM data were collected at the cryo-EM Swedish National Facility (funded by KAW, EPS, and Kempe foundations) at SciLifeLab. The mass spectrometry analysis was performed at the Swedish Metabolomics Centre, Umeå, Sweden (https://www.swedishmetabolomicscentre.se/) by Annika Johansson. This work was supported by the Swedish Foundation for Strategic Research (FFL15:0325), Ragnar Söderberg Foundation (M44/16), European Research Council (ERC-2018-StG-805230), Knut and Alice Wallenberg Foundation (2018.0080), Karolinska Institute and Max Planck Institute, H2020-MSCA-ITN-2016 (VS), H2020-MSCA-IF-2017 (YI), EMBO long-term fellowship LTF-2020-606 (MDN). We thank the Reviewers for their constructive comments and valuable suggestions.

## Additional information

### Funding

| Funder | Grant reference number | Author |
|---|---|---|
| Ragnar Söderbergs stiftelse | M44/16 | Alexey Amunts |
| European Research Council | ERC-2018-StG-805230 | Alexey Amunts |
| Knut och Alice Wallenbergs Stiftelse | 2018.0080 | Alexey Amunts Joanna Rorbach |
| EMBO | Long-term fellowship LTF-2020-606 | Minh Duc Nguyen |
| Karolinska Institutet | | Joanna Rorbach |
| Max Planck Institute for Biology of Ageing | | Joanna Rorbach |
| H2020-MSCA-IF-2017 | 799399-Itohribo | Yuzuru Itoh |
| H2020-MSCA-ITN-2016 | | Vivek Singh |

The funders had no role in study design, data collection, and interpretation, or the decision to submit the work for publication.

### Author contributions

Yuzuru Itoh, Conceptualization, Data curation, Formal analysis, Validation, Investigation, Writing – review and editing; Vivek Singh, Formal analysis, Validation, Visualization, Writing – review and editing; Anas Khawaja, Data curation, Visualization, Writing – review and editing; Andreas Naschberger, Formal analysis, Visualization, Writing – review and editing; Minh Duc Nguyen, Investigation; Joanna Rorbach, Conceptualization, Supervision, Funding acquisition, Writing – review and editing; Alexey Amunts, Conceptualization, Formal analysis, Supervision, Funding acquisition, Investigation, Writing - original draft, Project administration, Writing – review and editing

### Author ORCIDs

Vivek Singh http://orcid.org/0000-0003-4656-3362
Anas Khawaja http://orcid.org/0000-0002-9721-7454
Minh Duc Nguyen http://orcid.org/0000-0003-2945-9707
Joanna Rorbach http://orcid.org/0000-0002-2891-2840
Alexey Amunts http://orcid.org/0000-0002-5302-1740

### Decision letter and Author response

Decision letter https://doi.org/10.7554/eLife.77460.sa1
Author response https://doi.org/10.7554/eLife.77460.sa2

## Additional files

### Supplementary files
• MDAR checklist

### Data availability

The cryo-EM density maps and atomic coordinates have been deposited in the Electron Microscopy Data Bank (EMDB) and Protein Data Bank (PDB) under accession codes EMD-13170, EMD-15542, and 7P2E.

The following datasets were generated:

| Author(s) | Year | Dataset title | Dataset URL | Database and Identifier |
|---|---|---|---|---|
| Itoh Y, Khawaja A, Rorbach J, Amunts A | 2022 | Human mitochondrial ribosome small subunit in complex with IF3, GMPPMP and streptomycin | https://www.ebi.ac.uk/emdb/EMD-13170 | EMDB, EMD-13170 |
| Itoh Y, Khawaja A, Rorbach J, Amunts A | 2022 | Human mitochondrial ribosome small subunit in complex with IF3, GMPPMP and streptomycin | https://www.rcsb.org/structure/7P2E | RCSB Protein Data Bank, 7P2E |
| Itoh Y, Singh V, Khawaja A, Rorbach J, Amunts A | 2022 | Human mitochondrial ribosome small subunit in complex with streptomycin | https://www.ebi.ac.uk/emdb/EMD-15542 | EMDB, EMD-15542 |

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
