## [Editor Report]

This manuscript describes high-resolution structures of the small subunit of the human mitochondrial ribosome that reveal for the first time a number of physiologically important small molecules (GTP and NAD+/NADH) and iron-sulfur clusters integrated into the small subunit architecture. In the two structures, the authors also describe interactions of the small subunit with the antibiotic streptomycin that reveal how this antibiotic may be chemically modified by the cellular environment. The chemical-level detail revealed by these structures lays a foundation for future efforts to understand the basis for mitochondrial function in human health and disease.

---

## [Decision Letter]

**Decision letter after peer review:**

Thank you for submitting your article "Structural basis of streptomycin off-target binding to human mitoribosome" for consideration by *eLife*. Your article has been reviewed by 3 peer reviewers, one of whom is a member of our Board of Reviewing Editors, and the evaluation has been overseen by David Ron as the Senior Editor. The following individual involved in the review of your submission has agreed to reveal their identity: Tristan I Croll (Reviewer #2).

Essential revisions:

1) In the manuscript, the authors present both a streptomycin-bound mitochondrial small subunit as well as an apo-subunit structure (Figure 1c). However, there is no description of the structural determination for the unbound state in either the methods or Table 1. The authors should add this information. This also raises the question of whether the short format captures all that these structures can provide. Considering the size of these complexes and the dramatically improved resolution, surely there must be some other observations worthy of mention.

2) There are some questions about the interpretation of the density for the streptose ring. Although the overall density strongly supports the presence of bound streptomycin, the map does not convincingly support the proposed hydrated aldehyde within the streptomycin streptose moiety. While the claim that the poor density may reflect increased mobility of the hydrated aldehyde as the authors suggest is completely reasonable (and is supported by previously published NMR data), this would need to be independently validated given the ambiguity in the map. The authors could also cite two high-resolution crystal structures containing streptomycin (6fux and 5luh), which both show strong extra density consistent with the gem-diol form. A structure of streptomycin added in vitro to the mitochondrial small subunit would resolve whether the added molecule is indeed modified by the cellular milieu. While the authors suggest that adding the streptomycin to cultured cells rather than in vitro is an advantage of the work, the disadvantage is that it generates uncertainty that has not been resolved by the cryo-EM map alone about the chemical nature of the bound compound. The density might also be consistent with a potassium ion, for example.

A second issue with the streptose ring is the density surrounding the methyl group. The map is of high enough quality to suggest that it is possible that the stereochemistry for the methyl group has been inverted, possibly by an enzymatic activity within the cells. However, the map is not unambiguous in this regard. The reviewers think the authors would be justified in noting the possible change in stereochemistry as an alternative interpretation of the density.

3) The precise source of the streptomycin used in the study is not currently stated in the text. It is not clear if the compound was subjected to detailed mass spectrometry analysis to validate the composition assumed by the authors and to identify any potential chemical heterogeneity. This is an important control given the unexpected findings with the streptose ring in the structure.

Below are other itemized questions and comments from the reviewers for your consideration:*Reviewer #1 (Recommendations for the authors):*

1. Density for the O6" supports modeling two rotamers (Figure 1d, left), one like in the published structure, and the second as modeled here.

2. The modeling in Figure 1d to the right is really hard to follow, the way the figure is laid out.

3. The authors should show the density for the K71 interaction, as it is quite clear in the map, and would give the reader a sense of the map quality, with respect to the interactions.

4. Line 73 p. 3, Instead of shoulder, use body-shoulder junction?

*Reviewer #2 (Recommendations for the authors):*

– In the introduction, it is mentioned that mutation of C1494 to T increases susceptibility to streptomycin toxicity. The very strong excess density seeming to extend the base of A1492 (and to a lesser extent C1493) therefore seems to me to be in need of discussion (and ideally explanation). Given the spectacular local resolution it doesn't seem to be a simple artifact (and I note that there is more-or-less identical density associated with the same base in 6rw4) – is there any known evidence for modification at this base? I considered the possibility of an alternate conformation of the adjacent A1558, but (a) this clearly didn't fit the density, and (b) the excess density appears more contiguous with A1492 than with the complementary strand. I think some more careful evaluation of this site would not go astray.

– Much clearer in diagnosis, but less clear in terms of relevance: the base C931 (modelled stacking against Tyr31 of chain N) has a very clear alternate configuration that instead stacks against A1137 (a roughly 150 degree flip around the backbone). Considering that it is reasonably close (~25Å) to the bound streptomycin there may be some significance to this, but of course that is very difficult to speculate on.

– I found a small handful of what I believe are incorrect rotamers (Thr 4:492, His X:154, His H:147, His R:247, Arg N73). None of these are critical to interpretation of the model, but if you're feeling enthusiastic it might be worth revisiting them.

– Figure 1d: I think colour assignments in the caption are wrong (looks like this model is green, previous models are purple and cyan)?

– Page 9, line 203: "phenix.real_space_refine", not "phenix.real_space_refinement"

– Various small language issues (e.g. on line 19: "Inhibition of the human mitoribosome…"; line 24: "we report a 2.23Å resolution structure…"; etc.). None of this actually affects interpretation, in my opinion.*Reviewer #3 (Recommendations for the authors):*

1. It would help the reader if the authors could highlight in the main text what they consider are the key methodological innovations in their study that allowed them to generate such a high resolution map of the mitochondrial small subunit compared to the previously published studies.

2. Please clarify the source of the streptomycin used and what quality control assessment was made for the molecule.

3. To support the claim that a chemical modification has been introduced into the exogenously added streptomycin with the cell, the authors should either repeat the analysis using in vitro added streptomycin or try to obtain mass spectrometry data comparing the bound versus added form of the streptomycin.

4. Please improve the quality of Figure 1 to allow the reader to clearly distinguish the current model from the bacterial complexes. I find the presentation confusing. For example, the pink color is unexplained in the legend. On the left panel, which one is the current structure (green and cyan are stated to be bacterial in the legend)?

5. At least some discussion of the rest of the mitochondrial small subunit structure would be welcomed given the high resolution of the current mitochondrial small subunit map.

6. Line 86 "the previously modeled aldehyde"-modeled by whom? Please add a reference.

7. Does the structure explain why "patients carrying mtDNA mutations in the 12S rRNA gene, such as 1555A>G or 1494C>T more prone to aminoglycoside-induced ototoxicity" (line 49)? Please discuss.

8. Line 20: "Inhibition of human mitoribosome can be caused by antimicrobial off-target binding, which leads to clinical appearances". This is an example illustrating how the text would benefit from better clarity in the writing.

9. Given the high quality of the map, it is surprising that there is no discussion about the remainder of the mitochondrial small subunit, including ligands/spermine listed in Table 1. What is the rationale for GMPPNP and ATP, which are listed as ligands in Table 1?

---

## [Author Response]

Essential revisions:1) In the manuscript, the authors present both a streptomycin-bound mitochondrial small subunit as well as an apo-subunit structure (Figure 1c). However, there is no description of the structural determination for the unbound state in either the methods or Table 1. The authors should add this information. This also raises the question of whether the short format captures all that these structures can provide. Considering the size of these complexes and the dramatically improved resolution, surely there must be some other observations worthy of mention.

We followed the suggestion and added a structural analysis of the SSU that includes detection of previously unknown cofactor components of the mitoribosome and modifications. The analysis is outlined on page 3, lines 90-98, and described in 4 newly added sections.

Particularly, it includes the following content. The section ‘Protein modifications identified from the density’ (lines 106-129) reports mainchain modification 1-methyl-isoaspartate at position 184 in uS11m, and the N-terminal acetylations in bS21m and mS37.

The section ‘Iron-sulfur cluster involved in interactions between mitoribosomal proteins’ (lines 131-171) reports two iron-sulfur clusters that are coordinated by protein regions involving mitochondria-specific elements, and their positions are correlated with a loss of rRNA in the SSU.

The section ‘New GTP-binding site in the head of the mitoribosomal small subunit’ (lines 172-199) shows re-assignment of the previously incorrectly modeled GDP to ATP with an Mg^2+^ ion that allowed to identify a new GTP/GDP binding pocket.

The section ‘Spermine and NAD are associated with a single rRNA nucleotide insertion’ (lines 201-218) reports a spermine and NAD buried within the reduced rRNA core, bound to an insertion of the nucleotide C1048 in the rRNA.

The structural data is presented in new Figures 1 and 2. The title has also been changed to better reflect the added content to ‘Structure of the mitoribosomal small subunit reveals iron-sulfur clusters, a new GTP binding site, and basis for streptomycin off-target binding’

2) There are some questions about the interpretation of the density for the streptose ring. Although the overall density strongly supports the presence of bound streptomycin, the map does not convincingly support the proposed hydrated aldehyde within the streptomycin streptose moiety. While the claim that the poor density may reflect increased mobility of the hydrated aldehyde as the authors suggest is completely reasonable (and is supported by previously published NMR data), this would need to be independently validated given the ambiguity in the map. The authors could also cite two high-resolution crystal structures containing streptomycin (6fux and 5luh), which both show strong extra density consistent with the gem-diol form. A structure of streptomycin added in vitro to the mitochondrial small subunit would resolve whether the added molecule is indeed modified by the cellular milieu. While the authors suggest that adding the streptomycin to cultured cells rather than in vitro is an advantage of the work, the disadvantage is that it generates uncertainty that has not been resolved by the cryo-EM map alone about the chemical nature of the bound compound. The density might also be consistent with a potassium ion, for example.

As requested, we performed cryo-EM analysis of the in vitro bound streptomycin, as suggested. For this, the purified SSU (150 nM) was incubated with streptomycin solution (~600 µM) (Thermo Fisher, 15140122) for 30 min at room temperature to form the complex, before the sample was applied to the carbon coated grids for cryo-EM analysis. The resulting structure of the SSU with in vitro bound streptomycin was determined at 2.3 Å nominal resolution. As suggested by Reviewers, the streptose moiety is resolved better than in the native structure. Particularly, the density around the aldehyde moiety is branched and supports the modeling of one stabilized rotamer with the oxygen atoms within hydrogen bond distance from backbone phosphates of rRNA residues G898, G899, A1166 and A1167 and an adjacent water molecule. This indicates that the aldehyde is in the hydrated gem-diol state and participates in the potential interactions.

A section presenting the new data has been added on lines 255-282 and in Figure 3D. It has been further expanded in the Discussion section on page 12.

The cryo-EM of the in vitro bound streptomycin is shown in the new Figure 1—figure supplement 2.

The comparison with X-ray electron density maps of the streptomycin bound to adenylyltransferase AadA (PDB: 5LUH, Stern et al., 2018) and an aminoglycoside phosphotransferase APH(3″)-Id (PDB: 6FUX, Alekseeva et al., 2019) is discussed on lines 245-251 and shown in Figure 5—figure supplement 2.

A second issue with the streptose ring is the density surrounding the methyl group. The map is of high enough quality to suggest that it is possible that the stereochemistry for the methyl group has been inverted, possibly by an enzymatic activity within the cells. However, the map is not unambiguous in this regard. The reviewers think the authors would be justified in noting the possible change in stereochemistry as an alternative interpretation of the density.

Indeed, the results of the in vitro experiment showed no chirality flip around the methyl group, suggesting that the inversion of chirality of C42, where the methyl group of streptose is located, is affected by the cellular milieu in the native structure. This information has now been added on page 9, lines 236-239: “The density surrounding the methyl group also suggests the stereochemistry for the methyl group might have been inverted, possibly by enzymatic activity within cells, however the map quality is not unambiguous in this region”; on lines 263-264: “Unlike in the native structure, the streptose moiety is well resolved, and no chirality flip is observed around the methyl group”; and the data is shown in Figure 3D.

3) The precise source of the streptomycin used in the study is not currently stated in the text. It is not clear if the compound was subjected to detailed mass spectrometry analysis to validate the composition assumed by the authors and to identify any potential chemical heterogeneity. This is an important control given the unexpected findings with the streptose ring in the structure.

To confirm the chemical structure of the streptomycin, we now also performed mass-spectrometry analysis using a dual jet stream electrospray ion source operating in positive ion mode. The data is presented in Figure 5—figure supplement 3 and shows that the mass to charge ratio is consistent with the cryo-EM analysis of in vitro bound streptomycin displaying the highest peak at ~600.28, which corresponds to the hydrated gem-diol form, and a less-abundant peak at ~582.2, which corresponds to the exact m/z of streptomycin. This suggests that both forms of soluble streptomycin, hydrated and non-hydrated are retained in solution. The results are now described on page 10, lines 271-278, and the corresponding Methods section is on page 26, lines 569-594.

Reviewer #1 (Recommendations for the authors):1. Density for the O6" supports modeling two rotamers (Figure 1d, left), one like in the published structure, and the second as modeled here.

Thank you, we have updated the model to include the two alternative rotamers. Figure 3E shows the two alternative conformations.

2. The modeling in Figure 1d to the right is really hard to follow, the way the figure is laid out.

We have revised the panel (now Figure 3E) by removing one of the bacterial models and focusing on two separate elements in two individual panels. Thus, the middle panel shows the overview superposition of our structure with the previous model of streptomycin-bound T. thermophilus ribosome (Demirci et al., 2013, PDBID: 4DR3). The left panel has been replaced with a zoom-in to show alternative conformations of O6" N-methyl-L-glucosamine. The right panel shows a zoom-in to the methyl with its chemically more favorable conformation with the amino group at the 2″ position (protonated secondary amine), forming two hydrogen bonds/salt bridges with two backbone phosphates of rRNA. We hope the new illustration clarifies the data and also resolves discrepancies of the methyl-group orientation in the previous study.

3. The authors should show the density for the K71 interaction, as it is quite clear in the map, and would give the reader a sense of the map quality, with respect to the interactions.

We added Figure 5—figure supplement 1 that shows the densities of all the elements involved in streptomycin interactions, including the requested K71. In this figure, the bound streptomycin is shown as a model, and the interacting SSU region that forms the binding pocket is shown as a model with density.

4. Line 73 p. 3, Instead of shoulder, use body-shoulder junction?

We removed it to avoid confusion.

Reviewer #2 (Recommendations for the authors):– In the introduction, it is mentioned that mutation of C1494 to T increases susceptibility to streptomycin toxicity. The very strong excess density seeming to extend the base of A1492 (and to a lesser extent C1493) therefore seems to me to be in need of discussion (and ideally explanation). Given the spectacular local resolution it doesn't seem to be a simple artifact (and I note that there is more-or-less identical density associated with the same base in 6rw4) – is there any known evidence for modification at this base? I considered the possibility of an alternate conformation of the adjacent A1558, but (a) this clearly didn't fit the density, and (b) the excess density appears more contiguous with A1492 than with the complementary strand. I think some more careful evaluation of this site would not go astray.

We performed a more careful evaluation of this site and found two alternative states in this region, which is described on lines 297-304 and in the new Figure 5—figure supplement 4. Particularly, we found that A1558 is flipped out in one state, whereas its base is stacked between G1559 and A1492 in the other state, thus causing a shift of the residues in the other strand. Since this feature indicates a flexibility that might be related to recruitment of mRNA for translation, a possible mechanistic insight from our work is that by directly interacting with A1555 and C1556 on the mitoribosome, the streptomycin binding decreases that flexibility, and thus results in an enhanced translation inhibition.

– Much clearer in diagnosis, but less clear in terms of relevance: the base C931 (modelled stacking against Tyr31 of chain N) has a very clear alternate configuration that instead stacks against A1137 (a roughly 150 degree flip around the backbone). Considering that it is reasonably close (~25Å ) to the bound streptomycin there may be some significance to this, but of course that is very difficult to speculate on.

We thank the reviewer for pointing out this alternative conformation. We have revised the structure accordingly. In addition, we also performed a more careful analysis of the rest of the structure and identified alternative conformations for the following residues (which have also been introduced): chain-A G902, chain-B Met175, Ile232, chain-E Arg45, chain-Q Arg50, chain-S Ala2-Ser4, chain-T Arg160, chain-X Lys298, chain-Y Gln295. We have updated the deposited model accordingly and listed the alternative conformations in the Methods section ‘Model building and refinement’ on page 25, lines 558-560.

– I found a small handful of what I believe are incorrect rotamers (Thr 4:492, His X:154, His H:147, His R:247, Arg N73). None of these are critical to interpretation of the model, but if you're feeling enthusiastic it might be worth revisiting them.

Thank you, the rotamers have been fixed as suggested.

– Figure 1d: I think colour assignments in the caption are wrong (looks like this model is green, previous models are purple and cyan)?

The figure (now Figure 3) has been revised and the colour assignments have been corrected. It shows in red the density of the in vitro formed complex; on the left, zoom-in of the native state with blue density showing that the gem-diol moiety is not well resolved; on the right, zoom-in of the in vitro complex shows that the streptose moiety is better resolved, and the inversion of chirality of C42 is shown as well.

– Page 9, line 203: "phenix.real_space_refine", not "phenix.real_space_refinement"

Corrected.

– Various small language issues (e.g. on line 19: "Inhibition of the human mitoribosome…"; line 24: "we report a 2.23Å resolution structure…"; etc.). None of this actually affects interpretation, in my opinion.

Corrected.

Reviewer #3 (Recommendations for the authors):1. It would help the reader if the authors could highlight in the main text what they consider are the key methodological innovations in their study that allowed them to generate such a high resolution map of the mitochondrial small subunit compared to the previously published studies.

We added a section ‘Structure determination’ on page 1, where the requested methodological information is now described on lines 76-86. Particularly, we used 3D auto-refinement and 3D classification with local angular search with a solvent mask to remove poorly aligned particles. Then, the resolution was further improved by applying CTF refinement with beam-tilt, per-particle defocus, and per-micrograph astigmatism, followed by Bayesian polishing. Next, particles were separated into multi-optics groups. A second round of CTF refinement included beam-tilt, trefoil and fourth-order aberrations, magnification anisotropy, per-particle defocus, per-micrograph astigmatism. At the final stage, local-masked 3D auto-refinements were applied.

2. Please clarify the source of the streptomycin used and what quality control assessment was made for the molecule.

We added the requested description in Methods on lines 439-440:

“Penicillin-Streptomycin solution at pH 6.2 and osmolality 322 (Thermo Fisher, 15140122) was supplemented to the media with the final concentration of 100 units/mL of penicillin and 100 µg/mL streptomycin.”

For quality control assessment, we now performed High-resolution mass spectrometry (HRMS) analysis for streptomycin that is described in Methods on lines 571-596, and the results are reported in Figure 5—figure supplement 3.

3. To support the claim that a chemical modification has been introduced into the exogenously added streptomycin with the cell, the authors should either repeat the analysis using in vitro added streptomycin or try to obtain mass spectrometry data comparing the bound versus added form of the streptomycin.

Both had been performed, and the results with figures and discussion were added to the revised version of the manuscript. The details are described in our response to the Major points #2 (cryo-EM) and #3 (mass-spec) above.

4. Please improve the quality of Figure 1 to allow the reader to clearly distinguish the current model from the bacterial complexes. I find the presentation confusing. For example, the pink color is unexplained in the legend. On the left panel, which one is the current structure (green and cyan are stated to be bacterial in the legend)?

The quality has been improved in the revised version, and the presentation has been clarified. The details are described in our response to comment #5 from Reviewer 1 that is related to the same issue.

5. At least some discussion of the rest of the mitochondrial small subunit structure would be welcomed given the high resolution of the current mitochondrial small subunit map.

We added the requested analysis and discussion, as described in the response to the Major point #1 above.

6. Line 86 "the previously modeled aldehyde"-modeled by whom? Please add a reference.

References have been added to the X-ray crystal structures Carter et al., 2000; Demirci et al., 2013 (line 233).

7. Does the structure explain why "patients carrying mtDNA mutations in the 12S rRNA gene, such as 1555A>G or 1494C>T more prone to aminoglycoside-induced ototoxicity" (line 49)? Please discuss.

Thank you for encouraging us to expand on the medically related theme, as this aspect of the study is indeed important. We added an analysis of the structural basis for streptomycin toxicity with respect to the mutations 1494C>T and 1555A>G in h44 of the rRNA on lines 292-304. Particularly, the human mitochondrial rRNA appears to be more flexible in this region due to a loss of the G:C base-pair at the 1494:1555 position. The mutations then introduce Watson-Crick base pairs that limit the rRNA local flexibility and resemble the corresponding binding site in bacteria. Our density indicates two alternative states in this region, and previously this feature was reported in the context of recruitment of mRNA for translation (Khawaja et al., 2020). Thus, a possible mechanistic insight from our work is that by directly interacting with A1555 and C1556 on the mitoribosome, the streptomycin binding decreases that flexibility, and thus results in an enhanced translation inhibition.

8. Line 20: "Inhibition of human mitoribosome can be caused by antimicrobial off-target binding, which leads to clinical appearances". This is an example illustrating how the text would benefit from better clarity in the writing.

Replaced with “Inhibition of the human mitoribosome can be caused by antimicrobial drug off-target binding.”

9. Given the high quality of the map, it is surprising that there is no discussion about the remainder of the mitochondrial small subunit, including ligands/spermine listed in Table 1. What is the rationale for GMPPNP and ATP, which are listed as ligands in Table 1?

All have been discussed and illustrated in detail in the revised version on pages 4-8 with two new main figures, which is further reflected in the new title as well.